

# A large pterosaur limb bone from the Kaiparowits Formation (late Campanian) of Grand Staircase-Escalante National Monument, Utah, USA

Andrew A. Farke

Raymond M. Alf Museum of Paleontology at The Webb Schools, Claremont, CA, USA

## ABSTRACT

Pterosaurs were widespread during the Late Cretaceous, but their fossils are comparatively rare in terrestrial depositional environments. A large pterosaur bone from the Kaiparowits Formation (late Campanian, ~76–74 Ma) of southern Utah, USA, is tentatively identified as an ulna, although its phylogenetic placement cannot be precisely constrained beyond Pterosauria. The element measures over 36 cm in preserved maximum length, indicating a comparatively large individual with an estimated wingspan between 4.3 and 5.9 m, the largest pterosaur yet reported from the Kaiparowits Formation. This size estimate places the individual at approximately the same wingspan as the holotype for *Cryodrakon boreas* from the penecontemporaneous Dinosaur Park Formation of Alberta. Thus, relatively large pterosaurs occurred in terrestrial ecosystems in both the northern and southern parts of Laramidia (western North America) during the late Campanian.

## INTRODUCTION

Pterosaurs were a widespread component of terrestrial ecosystems during the Late Cretaceous, reconstructed as filling a variety of ecological niches (*Barrett et al., 2008*; *Witton & Naish, 2008*). However, the comparative rarity of skeletal material in most formations, due in part to strong taphonomic influences and other geological biases, have limited studies of this clade and clouded interpretations of pterosaur paleobiology and paleoecology (*Butler et al., 2012*; *Butler, Benson & Barrett, 2013*; *Dean, Mannion & Butler, 2016*). Thus, even isolated and incomplete bones can provide important information for establishing the distribution and general morphological attributes of pterosaurs (*Kellner et al., 2019*).

The Kaiparowits Formation preserves rocks deposited along the eastern margin of Laramidia during the late Campanian (~76.6–74.5 Ma; *Roberts, Deino & Chan, 2005*; *Roberts et al., 2013*), with significant exposures within Grand Staircase-Escalante National Monument (GSENM) in southern Utah. The formation represents a continental depositional environment preserving river channels, floodplains and associated settings. A rich fossil record includes exquisitely preserved specimens for numerous tetrapods,

Corresponding author
Andrew A. Farke, afarke@webb.org

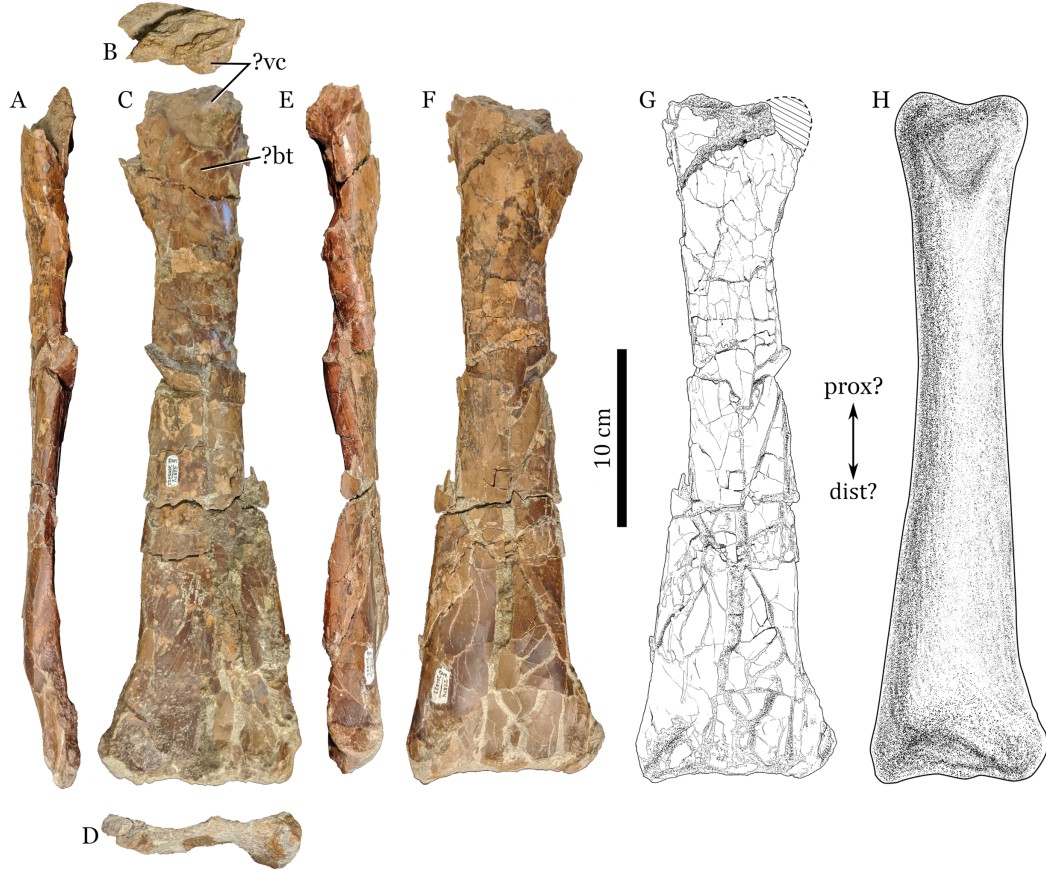

**Figure 1 RAM 22574, ulna of Pterosauria indet.** (A) ?dorsal; (B) ?proximal; (C) ?anterior; (D) ?distal; (E) ?ventral and (F) ?posterior views; with (G) showing interpretive drawing of ?posterior view, including missing parts; and (H) showing restored view of bone in ?posterior view. Scale bars equal 10 cm. Abbreviations: ?bt, ?bicipital tuberosity; ?dist, ?distal end; ?prox, ?proximal end; ?vc, ?ventral cotylus.

including birds, non-avian dinosaurs, crocodylomorphs, turtles, mammals, amphibians, and lepidosaurs (see *Titus & Loewen, 2013*, and references therein). Pterosaurs are known from only a handful of specimens. An isolated manual phalanx was the first published record of a pterosaur from the Kaiparowits Formation (*Farke & Wilridge, 2013*), suggesting a fairly small (<3 m wingspan) individual. A potential pteranodontoid metacarpal was later reported (*McCormack & Sertich, 2016*), as well as the incomplete but associated skeleton of an azhdarchid (*Carroll et al., 2017*), both of which await formal description.

Here I report on RAM 22574 (Fig. 1), an isolated ?ulna from the largest pterosaur (4.3–5.9 m estimated wingspan) yet known from the Kaiparowits Formation. Although it is only a single bone, this element establishes the size range occupied by the clade in southern Utah during the Late Cretaceous, and expands the known distribution of large pterosaurs across terrestrial environments during the late Campanian of western North America (Fig. 2).

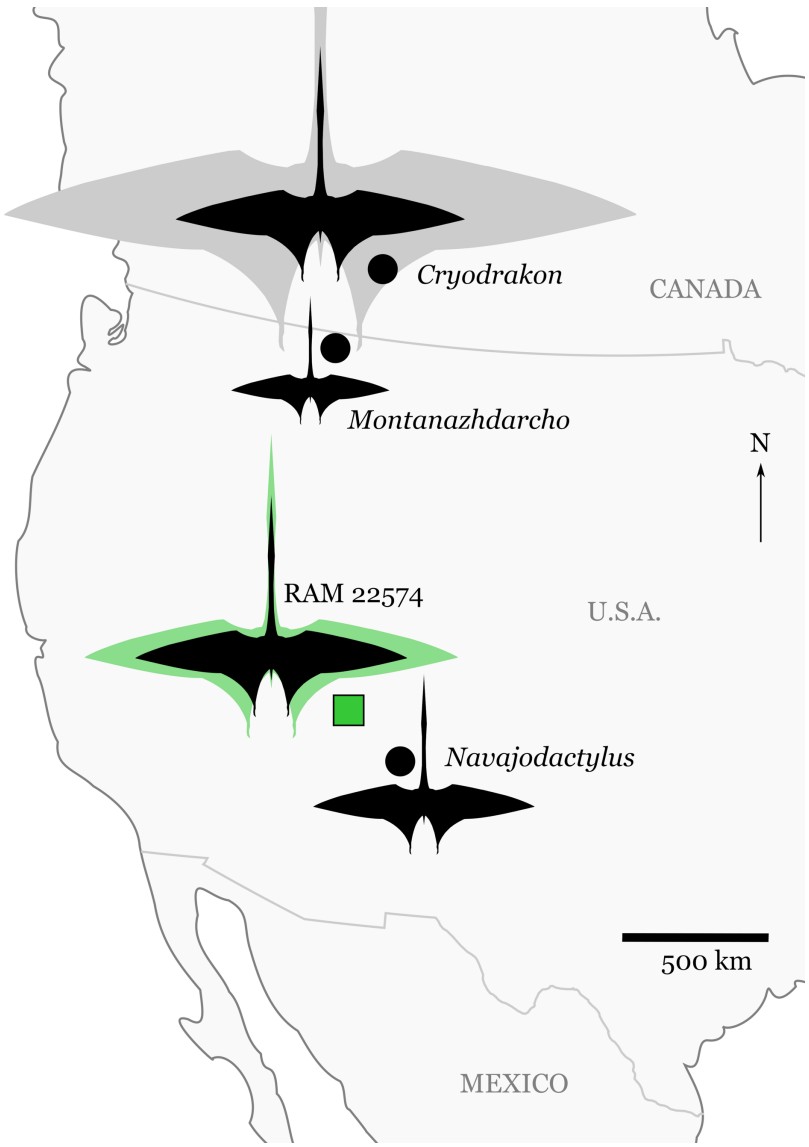

**Figure 2 Significant discoveries of pterosaurs from late Campanian-aged terrestrial depositional environments in western North America.** Silhouettes are scaled to maximum estimates of wingspan for individual specimens (see "Discussion" and Table 2). The silhouette for RAM 22574 shows the minimum (black) and maximum (green) size estimates for the specimen (Table 2). *Cryodrakon boreas* is scaled to the holotype (black), with the 10 m wingspan estimate (gray) for the largest known specimen (TMP 1980.16.1367) patterned after *Quetzalcoatlus northropi*. The silhouette is modified from an image of *Quetzalcoatlus* by Mark P. Witton and Darren Naish (licensed under CC BY 3.0 Unported via http://phylopic.org/); note that body shape may have varied greatly across the taxa depicted here.

## GEOLOGICAL SETTING

RAM 22574 was collected at locality RAM V2005022 (colloquially known as the "Cripe Site"), within the middle unit of the Kaiparowits Formation. This site is a multi-taxon bonebed including multiple associated elements from a tyrannosaurid, at least two
hadrosaurids, testudines, and a small (~3 m wingspan) azhdarchid pterosaur (*Farke et al., 2016*; *Carroll et al., 2017*).

The bonebed at RAM V2000522 measures over 1 m in thickness and is interpreted as including at least three main depositional events, with evidence of minor reworking of bones at the top of the sequence. RAM 22574 was collected at the stratigraphic top of the quarry, ~1.2 m above the lowest fossil, in a sandy mudstone with extensive clay rip-up clasts and plant debris. The articular surfaces of the bone show some pre-depositional damage, potentially due to fluvial abrasion or decomposition. The fossil was found within 1 m of an azhdarchid associated skeleton (RAM 15445), partially in the same bedding plane. However, RAM 22574 is from a much larger individual, in that its ulna measures 36 cm long, versus a radius length (which should be roughly equivalent to ulna length) of around 20 cm for RAM 15445.

## MATERIALS AND METHODS

### Collection and preparation

RAM 22574 was collected during the summer of 2016, using standard paleontological excavation techniques. Observations in the field showed that it was damaged prior to burial (described in more detail below). Paraloid B-72 was used to stabilize the fossil in the field, and as a consolidant and glue in the preparation lab. The fossil was mechanically prepared using pneumatic hand tools of various sizes, with final preparation completed using dental picks and pin vices. Fieldwork was conducted under US Bureau of Land Management-Utah paleontology permits UT06-012E-GS and UT06-001S-GS, and the fossil is reposited at the Raymond M. Alf Museum of Paleontology at The Webb Schools, Claremont, CA, USA.

### Comparisons and measurements

Linear measurements of RAM 22574 were collected with a digital calipers, to the nearest millimeter or 0.1 mm (Fig. 3; Table 1). Circumference was measured to the nearest millimeter with a cloth measuring tape. For anatomical comparisons, casts of *Montanazhdarcho minor* (MOR 691) and *Quetzalcoatlus northropi* (TMM 41450) were compared directly with RAM 22574.

### Wingspan estimation

The wingspan of RAM 22574 was approximated by scaling from relatively complete wings of pterodactyloid pterosaurs. Here, wing length is calculated as the sum of all sequential forelimb long bone lengths (humerus, ulna, metacarpal IV, phalanges in digit IV, excluding the carpals). Wingspan is approximated by doubling wing length. As noted by *Hone & Benton (2007)*, this neglects the width of the torso, but that is offset in part by the flexion of the wings in life. Data (Table 1) were taken from measurements published by *Unwin, Lü & Bakhurina (2000)* and *Bennett (2001a)*. Assuming that RAM 22574 was an ulna, each wing was scaled by ulna size for that specimen. To reduce concerns about allometry, only specimens in the approximate size range of RAM 22574 were used (<25% difference in ulnar length). Because RAM 22574 was slightly "telescoped," two calculations

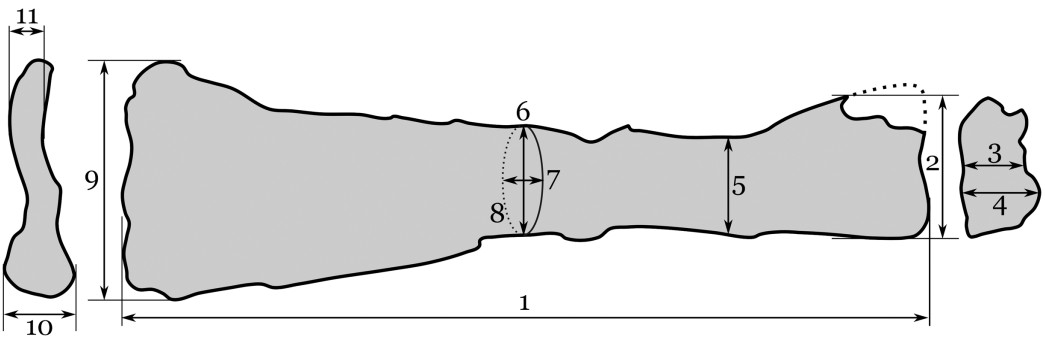

**Figure 3 Interpretive drawing of RAM 22574, showing measurements taken here.** Measurements include: (1a) maximum proximo-distal length; (1b) maximum proximo-distal length (adjusting for telescoping); (2) maximum dorso-ventral width of proximal end; (3) minimum antero-posterior width of proximal end; (4) antero-posterior width of proximal end at ventral cotyle; (5) dorso-ventral width at narrowest point of shaft; (6) dorso-ventral width at mid-shaft; (7) antero-posterior width at mid-shaft; (8) circumference at mid-shaft; (9) maximum dorso-ventral width of distal end; (10) maximum antero-posterior width of distal end; (11) minimum antero-posterior width of distal end. Data are provided in Table 1.                               

**Table 1 Measurements of pterosaur ?ulna, RAM 22574, in millimeters.** All measurements were taken with sliding digital calipers, except for 6, which was measured with a cloth measuring tape. See Fig. 3 for explanation of measurements.

|  | Standard | Measurement |
|---|---|---|
| 1a | Maximum proximo-distal length (as preserved) | 366 |
| 1b | Maximum proximo-distal length (adjusting for telescoping) | 381 |
| 2 | Maximum dorso-ventral width of proximal end (as preserved) | 71 |
| 3 | Minimum antero-posterior width of proximal end (as preserved) | 32 |
| 4 | Antero-posterior width of proximal end at ventral cotyle | 48 |
| 5 | Dorso-ventral width at narrowest point of shaft | 49 |
| 6 | Dorso-ventral width at mid-shaft (as preserved) | 52 |
| 7 | Antero-posterior width at mid-shaft (as preserved) | 27 |
| 8 | Circumference at mid-shaft (as preserved) | 133 |
| 9 | Maximum dorso-ventral width of distal end (as preserved) | 106 |
| 10 | Maximum antero-posterior width of distal end (as preserved) | 31 |
| 11 | Minimum antero-posterior width of distal end (as preserved) | 17 |

were run—one with the bone length as preserved, and another adding an additional 15 mm to the bone length to accommodate the effects of the crushing. Finally, the wingspan of *Cryodrakon boreas* (TMP 1992.83.4) was estimated by scaling the holotype humerus (measurements from *Godfrey & Currie, 2005*) relative to the humerus for *Quetzalcoatlus* sp. (TMM 42422).

# RESULTS

## Systematic Paleontology

Archosauria *Cope, 1869* sensu *Gauthier & Padian, 1985*

Pterosauria *Kaup, 1834* sensu *Sereno, 1991*

**Referred material.** RAM 22574, a nearly complete ?ulna (Fig. 1).

**Locality and horizon.** Locality RAM V2005022, the "Cripe Site," located within Grand Staircase-Escalante National Monument, Garfield County, Utah (Fig. 2). The site is situated in the middle unit of the Kaiparowits Formation, which is late Campanian in age (*Roberts, Deino & Chan, 2005*; *Roberts et al., 2013*).

**Identification.** This fossil is identified confidently as a pterosaur limb bone, and tentatively as an ulna. Because the descriptive terminology hinges upon these assumptions, I first address the underlying logic.

RAM 22574 is clearly hollow, which for the Late Cretaceous restricts possible identifications to either Theropoda or Pterosauria. The extremely thin cortical bone (between 0.7 and 1.7 mm) relative to the size of the element is distinct to pterosaurs versus their outgroup clades (*Unwin, 2003*), particularly for elements of this size. Thus, the identification to Pterosauria is quite confident.

Because key parts of RAM 22574 were damaged prior to fossilization (see below), identification of this bone within the skeleton is less certain. It is clearly a limb bone (rather than vertebra, ribs, or cranial material), but does not match well with morphology expected for any of the hind limb elements. There is nothing that resembles either the head or distal end of the femur, nor the somewhat sigmoid, relatively slender profile of a typical pterosaur femur. The overall robustness of the bone (proportions of length vs. width) differs sharply from that seen in the tibia and fibula for typical Late Cretaceous pterosaurs of this size range (*Bennett, 2001b*; *Averianov, 2010*). A humerus can be excluded on the basis of a lack of a deltopectoral crest or the bulbous distal articular surface processes. Neither articular end shapes or element proportions fall within what would be expected for metacarpals or phalanges, regardless of clade. Thus, a radius or ulna seems to be the most likely identification.

The more heavily pre-depositionally damaged end of RAM 22574 shows topographic complexity that differs from what is seen in typical pterosaur radii (*Wellnhofer, 1991*; *Bennett, 2001b*; *Veldmeijer, 2003*). A prominent protrusion (labeled as ?vc on Figs. 1B and 1C) resembles a similar expansion associated with the ventral cotylus seen in *Montanazhdarcho minor* (MOR 691, personal observation on cast; ulnar facet/fossa of *McGowen et al., 2002*) and also an ulna referred to *Cryodrakon boreas* (TMP 1965.14.398; *Godfrey & Currie, 2005*; *Hone, Habib & Therrien, 2019*). The relatively restricted nature of this protrusion differs from the more elongate, tab-like anterior tuberosity seen on a radius referred to *Azhdarcho lancicollis* (CCMGE 8/11915; see fig. 26 in *Averianov, 2010*).

The overall proportions (element length vs. maximum diaphyseal width) of RAM 22574 also are more similar to that of an ulna than a radius. For instance, *Quetzalcoatlus northropi* (TMM 41450) shows that radii tend to be far more slender (proportionately) than ulnae. Similar proportions are seen in *Montanazhdarcho minor*, *Azhdarcho lancicollis*, *Maaradactylus spielbergi*, and also in *Pteranodon* spp. (*Bennett, 2001b*; *McGowen et al., 2002*; *Veldmeijer, 2003*; *Averianov, 2010*), as a few examples. In the
*Maaradactylus spielbergi* holotype (RGM 401880; *Veldmeijer, 2003*), which has approximately the same ulnar length as the length of RAM 22574, the dorso-ventral diaphyseal diameter to maximum element length ratio is around 0.076 for the ulna and 0.033 for the radius. The same ratio is approximately 0.13 in RAM 22574 (see Table 1).

Assuming that RAM 22574 is an ulna, it can be tentatively identified as a right element, based on the position of a roughened area on the ?proximal end that may represent the bicipital tuberosity (labeled as ?bt on Fig. 1C). This area should be towards the ventral edge of the element's anterior surface, as seen in other pterosaurs (e.g., *Azhdarcho lancicollis*, ZIN PH 41/44; see fig. 25 in *Averianov, 2010*).

**Description.** RAM 22574 measures 366 mm in maximum preserved length. As outlined above, this description assumes that RAM 22574 is a right ulna, so that directional and anatomical terminology follow accordingly. Both proximal and distal ends were slightly abraded prior to fossilization, with the proximal end more severely damaged. Part of this end was broken open prior to burial, as evidenced by some rip-up clay clasts inside the bone as well as field observations of the incomplete element by the author. Little of the actual articular surface is preserved on either end. Surface bone texture for the shaft is smooth rather than porous, and no open epiphyses are visible, suggesting osteological maturity for this individual (although not definitively so).

The proximal end of the element is mostly incomplete, preserving only a portion of what is interpreted as the ventral cotylus. The cotylus projects anteriorly, with much of its articular surface abraded away. A roughened patch of bone around 20 mm distal to the peak of the cotylus may represent the bicipital tuberosity, for attachment of *m. biceps brachii* (Fig. 1C). There is no evidence of pneumatic foramina on the proximal end of RAM 22574, but the area where such foramina would be expected is broken.

The shaft broadens gradually from the proximal to the distal end of the element (Figs. 1C, 1F, and 1G), although this appears accentuated in part by crushing. At mid-shaft, the cross-section is oval and elongated in the dorso-ventral direction (width/height ratio of 0.51; see Table 1). At the distal third of the bone, cortical thickness ranges from 1.0 to 1.6 mm; at the proximal third, it is 1.7 mm thick. At least part of the shaft is mildly telescoped through proximo-distal distortion, as shown by displacement around the shaft. This would add another 15 mm or so to the total length of the bone. The distal third of the shaft in RAM 22574 is flattened on its ventral surface. This is somewhat accentuated by crushing, but appears to be at least partly original morphology.

The distal end of RAM 22574 is slightly abraded, but overall appears to be more intact than the proximal end. Its distal margin in anterior view is relatively straight, probably accentuated by abrasion. When viewed end on, the ventral condyle region is more expanded than the dorsal condyle, although it appears that the dorsal condyle is abraded (Fig. 1D). A broad depression separates the condyles on the posterior surface of the element, although the extent of this depression appears accentuated by crushing. After accounting for the space between the crushed bone pieces, the width of the distal end is exaggerated by around 10 percent.

**Table 2 Comparative measurements of selected pterodactyloid pterosaurs, with wingspan for RAM 22574 scaled from those measurements.** Two ulna lengths are provided for RAM 22574, representing the element as preserved (the smaller number) and a second estimate accounting for mild telescoping that reduced the preserved length of the element. Measurements for *Pteranodon* are taken from *Bennett (2001a)*, and those for *Cryodrakon* are from *Godfrey & Currie (2005)*; all others are from *Unwin, Lü & Bakhurina (2000)*. Measurements are in millimeters, except for wingspan estimates, which are in meters. Abbreviations: H, humerus length; MC-IV, metacarpal IV; IV-1,-2,-3,-4, fourth digit manual phalanges 1 through 4; RAM WS, range of wingspans estimated for RAM 22574 based on direct scaling from each specimen; WS, wingspan (calculated by summing forelimb bone lengths and multiplying by 2).

| Taxon | Specimen | H | U | MC-IV | IV-1 | IV-2 | IV-3 | IV-4 | WS | RAM WS |
|---|---|---|---|---|---|---|---|---|---|---|
| *Arthurdactylus conandoylei* | SMK 1132 PAL | 230 | 312 | 227 | 445 | 402 | 312 | 275 | 4.41 m | 5.2–5.4 m |
| *Anhanguera santanae* | NSM PV 19892 | 257 | 384 | 257 | 462 | 387 | 270 | 225 | 4.48 m | 4.3–4.5 m |
| *Cryodrakon boreas* | TMP 1992.83.4 | 245 | | | | | | | 4.57 m (est.) | |
| *Pteranodon* sp. | FHSM 184 | 269 | 393 | 583 | 653 | 539 | 390 | 194 | 6.04 m | 5.6–5.9 m |
| *Quetzalcoatlus* sp. | TMM 42422 | 250 | 358 | 620 | 602 | 305 | 156 | 39 | 4.66 m | 4.8–5.0 m |
| Pterosauria indet. | RAM 22574 | | 366/381 | | | | | | | 4.3–5.9 m |

## DISCUSSION

Unfortunately, the incomplete and crushed nature of RAM 22574 limits interpretation of the element and the animal. In general, neither the radius nor ulna exhibit major diagnostic features in pterosaurs, so the element cannot currently be identified beyond Pterosauria. Nevertheless, RAM 22574 does represent the largest pterosaur bone yet known from the Kaiparowits Formation and only the second formally described element, and is thus useful for documenting the size of some of the pterosaurs in the Kaiparowits ecosystem.

The total wingspan for RAM 22574 is estimated at 4.3–5.9 m (based on the highest and lowest estimates in Table 2). This places it within the same size range as specimens of *Quetzalcoatlus* sp. (TMM 42422; Maastrichtian, Javelina Formation, Texas) or *Cryodrakon boreas* (TMP 92.83.4; late Campanian, Dinosaur Park Formation, Alberta); the ulna of RAM 22574 is roughly the same length as that of TMM 42422. *McGowen et al. (2002)* estimated a 2.5 m wingspan for *Montanazhdarcho minor* (late Campanian, Two Medicine Formation, Montana) and *Sullivan & Fowler (2011)* estimated 3.5 m for *Navajodactylus* (late Campanian, Kirtland Formation, New Mexico). Thus, RAM 22574 stands alongside the holotype specimen of *C. boreas* as one of the larger pterosaur individuals known from late Campanian-aged terrestrial deposits of North America. Furthermore, it demonstrates that pterosaurs in this size range occurred in ecosystems both in the northern and southern parts of Laramidia (western North America) during the late Campanian (see Fig. 2). RAM 22574 is nonetheless from an individual smaller than the largest referred element of *C. boreas* (TMP 1980.16.1367), which may have matched some of the largest azhdarchids in size (*Hone, Habib & Therrien, 2019*). This indicates potential for truly gigantic pterosaurs (≥10 m wingspan) across terrestrial environments in North America. Future discoveries will undoubtedly help clarify phylogenetic relationships between pterosaurs living in terrestrial environments at this time, to see if they were relatively geographically restricted, or if individual species had continent-level ranges. Additionally, more work is required to determine if large pterosaurs played similar ecological roles across their various environments.

## ABBREVIATIONS

| | |
|---|---|
| **CCMGE** | Chernyshev's Central Museum of Geological Exploration, Saint Petersburg, Russia |
| **FHSM** | Sternberg Museum of Natural History, Hays, KS, USA |
| **NSM** | National Science Museum, Tokyo, Japan |
| **RAM** | Raymond M. Alf Museum of Paleontology at The Webb Schools, Claremont, CA, USA |
| **RGM** | Nationaal Natuurhistorisch Museum (Naturalis Biodiversity Center), Leiden, The Netherlands |
| **SMK** | Staatliches Museum für Naturkunde, Karlsruhe, Germany |
| **TMM** | Texas Memorial Museum, Austin, TX, USA |
| **TMP** | Royal Tyrrell Museum of Palaeontology, Drumheller, AB, Canada |
| **ZIN PH** | Paleoherpetological Collection, Zoological Institute of the Russian Academy of Sciences, Saint Petersburg, Russia |

## ACKNOWLEDGMENTS

This specimen would never have been revealed without the sharp eyes of Jeff Cripe, who discovered the locality where this bone was collected. The 2016 field crew of volunteers and museum staff from the Alf Museum assisted in collection of this fossil, and Jared Heuck skillfully completed preparation. Discussions with Dave Hone, Nate Carroll, and others were helpful in establishing and confirming the identity of this fossil. I thank Valeria Pellicer for her renderings of the bone. Fieldwork was facilitated by Alan Titus and Greg MacDonald. Editorial comments by Graciela Piñeiro and reviewer comments by Dave Hone, Felipe Pinheiro, and Sterling Nesbitt improved and clarified an earlier version of this manuscript. Finally, I thank Frank Varriale for being my "writing buddy" who helped push this project to completion.

### Funding

Funding for this research was provided by the David B. Jones Foundation, with additional support from the Augustyn Family. The funders had no role in study design, data collection and analysis, decision to publish, or preparation of the manuscript.

### Grant Disclosures

The following grant information was disclosed by the authors:
David B. Jones Foundation.

### Competing Interests

Andrew A. Farke is an Academic Editor and Section Editor for PeerJ.

## Author Contributions

- Andrew A. Farke conceived and designed the experiments, performed the experiments, analyzed the data, prepared figures and/or tables, authored or reviewed drafts of the paper, and approved the final draft.

## Field Study Permissions

The following information was supplied relating to field study approvals (i.e., approving body and any reference numbers):

Fieldwork in GSENM was completed under US Bureau of Land Management paleontology permits UT06-012E-GS and UT06-001S-GS.

## Data Availability

All relevant data are available in the Tables of the article itself.

The specimen described here, RAM 22574, is housed at the Raymond M. Alf Museum of Paleontology at The Webb Schools, located in Claremont, CA, USA.

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
