# Peer review of "A large pterosaur limb bone from the Kaiparowits Formation (late Campanian) of Grand Staircase-Escalante National Monument, Utah, USA"

_PeerJ, doi:10.7717/peerj.10766_

## Round 0.1 · original submission

Graciela Piñeiro · · Academic Editor
Dear author,
We have received three review reports about your article entitled “A large pterosaur limb bone from the Kaiparowits Formation (late Campanian) of Grand Staircase-Escalante National Monument, Utah, USA”. All the colleagues agreed that it would need just a few additional works for to be accepted for publication in PeerJ. Please consider all the suggestions and recommendations from the reviewers, including those marked in the annotated pdf kindly provided by reviewer 3.
In particular I agree with reviewer 3 that would be useful that you provide a stratigraphic section of the Kaiparowits Formation for indication of the level of the described specimen (RAM 22574) and a photograph showing the bone bed containing other pterosaur remains. Moreover, please include a hypothesis that explains the “movement” of the bones up to the top of the section, are you meaning that the bones suffered some kind of edafization?
Best regards,
Graciela Piñeiro
David Hone ·
Basic reporting
There are no fundamental problems with this paper. It is short and simple and there is little that is really debatable let alone problematic. There are however a fair number of points where the writing could be clarified and some additional small details or comparisons would make it rather easier to follow and better explain the author's position. All of the minor comments are presented on a marked-up document.
One thing that would help especially would be a simple figure (line drawing or photo) of the typical Cretaceous pterosaur radius and ulna to compare to the specimen.
Experimental design
This is all fine. Again, there are some areas that could be a little clearer.
Validity of the findings
Basically all fine. As above there are places where this can be more clearly expressed, especially in the lines about body size and overlap with other contemperaneous taxa.
Felipe Lima Pinheiro ·
Basic reporting
In this short communication, Dr. Farke presents a pterosaur ulna from the North American continental Kaiparowits Fm. Although incomplete and not that informative, the bone is worth publication because pterosaurs from this specific time interval (Campanian), environment (fluvial) and geography (western North America) are comparatively rare. The MS is written in plain, professional English, and I have just a single, very minor observation regarding the language. Dr. Farke’s work is self-contained, with all relevant data adequately provided within the text, figures and tables. The text is fully referenced, and the author was successful in providing an adequate background for his research.
Experimental design
As the studied specimen is not that informative, methods were restricted to morphological description and comparisons, what is adequate in this situation.
Validity of the findings
I agree with Dr. Farke’s identification of this bone as a right ulna and with the provided size estimations. Still, I believe that the author’s conclusions are too unpretentious and that this specimen welcomes a small piece of (well-grounded) speculation regarding the owner of the bone. In my understanding, pterosaurs from the Campanian are restricted to pteranodontids, nyctosaurids and azhdarchids. It is very unlikely that a nyctosaurid produced this bone: not only nyctosaurids were comparatively small animals, but they also had much more slender ulnae. The ulna of Pteranodon seems slightly less stout than RAM 22574, but pteranodontids with similar ulnar proportions do exist (e.g. Longrich et al., fig. 4). More interestingly, pteranodontids are indeed strongly correlated to coastal/marine environments (I know no exceptions). Accordingly, the size, faunistic context (azhdarchid remains were found in close association to the new specimen), and lithology (azhdarchids are more likely than not found in continental deposits) makes an indeterminate azhdarchid the most likely owner of this particular bone. Although speculative, this is apparently the most parsimonious hypothesis, and the author may want to acknowledge this in his text.
Additional comments
Geological Setting: Please indicate more clearly the depositional environment of the Kaiparowits Fm.
Lines 15-16: Azhdarchids seem to challenge this particular bias, with most records coming from inland deposits (see Witton and Naish, 2008).
Line 124: femora are also much more slender and slightly sigmoid, very different from what is displayed by RAM 22574.
Line 144: delete 'in'
Sterling Nesbitt ·
Basic reporting
The contribution is well written, the references are up to date and sample classic and recent publications, the hypothesis of identification is clear and the discussion/conclusions follow the data (the study is repeatable based on what is presented). The text is clean minus a typo here and there (see pdf)
Experimental design
The paper absolutely falls within the scope of the journal. All permits are cited and the methodology is detailed. It would be nice if the author provided a 3d model of the ulna if possible.
Validity of the findings
The conclusions are based on sound data and the author indicates where there could be interpretation differences. The implications are provided and are backed up by data.
Additional comments
This paper is nearly ready to go. Other than a few tiny items on the pdf, my biggest suggestion is to add a map and maybe a strat section that summarizes the geologic and geographic occurrence.

---

## Round 0.2

Graciela Piñeiro · · Academic Editor
Dear author,
Given that reviewers have accepted your explanation for not including part of their previous requests; I agree that your article is almost ready to be accepted for publication in PeerJ. However, Reviewer 3 has made some additional suggestions that will be easy for you to address. After that, please submit the revised version of your manuscript, surely ready for final approval.
With my best regard,
Graciela Piñeiro
Felipe Lima Pinheiro ·
Basic reporting
-
Experimental design
-
Validity of the findings
-
Additional comments
I'm happy with the final version of the MS.
Sterling Nesbitt ·
Basic reporting
This paper is ready to go with a few additional suggestions based on changes from the first draft. I think the author did an excellent job incorporating reviewer comments.
Experimental design
NA
Validity of the findings
Excellent as was before.
Additional comments
Minor comments:
Systematic Paleontology
-it would be helpful to provide the phylogenetic definition – e.g., Archosauria sensu Gauthier and Padian 1985
-I would like to reiterate you do not need the Pterosauria Genus and species indeterminate.
-it is redundant given that every individual belongs to a genus an a species
-it is also a vestige of Linnean taxonomy – I would be happy to chat about it if needs be
Line 254 – ‘truly gigantic pterosaurs’ please give a wing span range to slightly quantify
Figure 2 – the head of your large size of Cryodrakon is cut off, move down or over a bit?
-nit picky on my part, but I think your outline of the coasts can be less rounded (maybe create in a non-vector program) so that it matches the real coastline better.
-make the boundaries of the US (gray) have less weight because it clearly overlies the coast line and enters bodies of water
-given that you have political boundaries, it is necessary to have countries labeled for non-US researchers.
I am happy for this to be published without any more reviews from me

---

## Round 0.3 · accepted

Graciela Piñeiro · · Academic Editor
Dear author,
Thanks for having considered the last modifications suggested by the reviewers to improve your manuscript; it is now ready for publication in PeerJ. I think it will be an interesting paper for pterosaur specialists, contributing in particular to the better knowledge about the distribution of terrestrial giant species during the Late Campanian.
Congratulations!
Best regards,
Graciela Piñeiro